# Whole-Genome Sequencing of 502 Individuals from Latvia: The First Step towards a Population-Specific Reference of Genetic Variation

**DOI:** 10.3390/ijms242015345

**Published:** 2023-10-19

**Authors:** Raimonds Reščenko, Monta Brīvība, Ivanna Atava, Vita Rovīte, Raitis Pečulis, Ivars Silamiķelis, Laura Ansone, Kaspars Megnis, Līga Birzniece, Mārcis Leja, Liqin Xu, Xulian Shi, Yan Zhou, Andis Slaitas, Yong Hou, Jānis Kloviņš

**Affiliations:** 1Latvian Biomedical Research and Study Centre, LV-1067 Riga, Latvia; monta@biomed.lu.lv (M.B.); ivanna.atava@biomed.lu.lv (I.A.); vita.rovite@biomed.lu.lv (V.R.); raitis@biomed.lu.lv (R.P.); ivars.silamikelis@biomed.lu.lv (I.S.); laura.ansone@biomed.lu.lv (L.A.); kaspars.megnis@biomed.lu.lv (K.M.); liga.birzniece@biomed.lu.lv (L.B.); klovins@biomed.lu.lv (J.K.); 2Faculty of Medicine, University of Latvia, LV-1004 Riga, Latvia; cei@latnet.lv; 3Institute of Clinical and Preventive Medicine, University of Latvia, LV-1079 Riga, Latvia; 4Latvia MGI Tech, LV-2167 Mārupe, Latvia; xuliqin@mgi-tech.com (L.X.); shixulian@mgi-tech.com (X.S.); zhouyan1@mgi-tech.com (Y.Z.); andisslaitas@mgi-tech.com (A.S.); houyong@mgi-tech.com (Y.H.)

**Keywords:** whole-genome sequencing, genetic variation, population-specific reference panel, imputation performance, Latvian genomes

## Abstract

Despite rapid improvements in the accessibility of whole-genome sequencing (WGS), understanding the extent of human genetic variation is limited by the scarce availability of genome sequences from underrepresented populations. Developing the population-scale reference database of Latvian genetic variation may fill the gap in European genomes and improve human genomics research. In this study, we analysed a high-coverage WGS dataset comprising 502 individuals selected from the Genome Database of the Latvian Population. An assessment of variant type, location in the genome, function, medical relevance, and novelty was performed, and a population-specific imputation reference panel (IRP) was developed. We identified more than 18.2 million variants in total, of which 3.3% so far are not represented in gnomAD and dbSNP databases. Moreover, we observed a notable though distinct clustering of the Latvian cohort within the European subpopulations. Finally, our findings demonstrate the improved performance of imputation of variants using the Latvian population-specific reference panel in the Latvian population compared to established IRPs. In summary, our study provides the first WGS data for a regional reference genome that will serve as a resource for the development of precision medicine and complement the global genome dataset, improving the understanding of human genetic variation.

## 1. Introduction

Human population genetics benefitted from the completion of the human genome sequence [1], which was further advanced by creating the reference of global genome variation [2] and, finally, the establishment of regional references assessing fine details of local variation in whole-genome sequences. Although European populations are relatively well represented in this respect, compared to other parts of the world, in many countries the data on genetic variability are still lacking [3]. The population of Latvia is of particular interest, being the most distant European population from African and Asian clusters of principal component analysis (PCA) [4,5,6,7,8] and, together with neighbouring Baltic populations, exhibit relatively high ancestry proportion of two European founding populations, Western European hunter-gatherer and Yamnaya [6,9,10].

Genomic diversity in the population of Latvia is closely related to other European populations and was shaped by three principal migration events [6,9]. The first traces back to Western European hunter-gatherers (WHG), who expanded from the refuges of the last glacial maxima to the rest of the European continent. The Early European Farmer (EEF) migration from Anatolia started around 7000 BC and spread throughout mainly South-Central Europe with a modest contribution to the population of Latvia [6,10]. Lastly, beginning around 3000 BC, the Indo-European language speaking Western Steppe Herder (WSH) Yamnaya population spread rapidly from the Eurasian steppe throughout Europe, establishing the last major contribution to the underlying genetic makeup of Latvian and other European populations [6,9,10].

Currently, there are no studies representing detailed whole-genome variation in the Latvian population. Such a dataset would be beneficial for various applications, including the discovery of new medically relevant variations [11], assessing population-wide genetic risks [12], and improving genome imputation in related populations [13]. In the current study, we sequenced, analysed, and tested the whole-genome sequencing data of 502 individuals from the population of Latvia. We assessed the quality of sequences, selected confidently called variants, compared the accuracy to genotypes, performed population stratification, and annotated a diverse set of variants. We then imputed unrelated genotypes using a Latvian-population-specific imputation panel and compared its performance versus the 1000 Genomes Project and TOPMed imputation reference panels.

## 2. Results

### 2.1. Characteristics of the Cohort

Overall, 887 participants had their genomes sequenced and deposited in the Genome Database of the Latvian population (LGDB) in July 2022. Out of them, 295 sequences were not further processed due to missing data for structural variants and mobile element insertions, excess heterozygosity (n = 33), relatedness (n = 20), and the age threshold set for study participants < 18 years (n = 37). The regional distribution of the study participants according to their self-reported place of birth (which was restricted to indicating one of the five regions of Latvia without specifying the exact birthplace), self-reported ethnicity, as well as their age and sex, are provided in Table 1. The final study group consisted of 502 individuals denoted as LVBMC (Latvian Biomedical Research and Study centre) dataset. The majority (62%) were females and age of participants ranged from 18 to 91. All of the regions and most common ethnic groups of Latvia were well represented, with Latvians comprising the majority (62.5%) of the study participants.

### 2.2. Quality of Variant Calling

Variant-calling quality assessments were performed on all 502 samples included in the analysis, with an average sequencing depth of 35.7. The average read count was 9.15 × 10^8^ [Interquartile Range (IQR) = 7.4 × 10^8^–9.9 × 10^8^] and the average mapped read proportion was 99.86% [IQR = 99.98–99.99]. For each individual sample, we also assessed the read coverage distribution along the genome. On average, 89.71% [IQR = 89.19–90.68] of each genome reached at least 10× coverage, 63.09% [IQR = 52.98–73.62] reached 30×, and 19.11% [IQR = 6.94–27.64] reached at least 50× coverage.

A significant proportion of samples in this study (316 of 502) were genotyped using Illumina Array, allowing us to compare the accuracy of the variant calling between sequencing and genotyping platforms. Comparison of 489,918 overlapping variants revealed an overall concordance of 98.81%, with the most frequent discrepancies being the homozygous reference allele estimated using WGS, which was called either a heterozygote (52.6% of all discrepancies) or homozygous alternative allele (39% of all discrepancies) when called by genotyping. Additionally, we calculated an overall non-reference (NR) sensitivity of 0.96 and an NR discordancy rate of 0.5%.

### 2.3. Genetic Variation in the Latvian Population 

In total, 30,919,589 SNPs and INDELs were detected with an error rate of 0.1%. An allele count of three or more was observed for 18,266,684 variants, which were included in all subsequent analyses. In total, 3.3% of variants were not present in the gnomAD dataset (Table 2). The variant count ranged from 4.25 × 10^6^ to 4.72 × 10^6^ small polymorphisms per sample.

We managed to detect 61,605 large variants, including structural variants and mobile element insertions (Table 2). The rate of novel structural variants was an order of magnitude higher compared to small variants, with ALU (18.65%), LINE1 (26.7%), and non-autonomous retroelements SVA (23.7%) having the lowest proportion of novel variations. We observed considerably higher inter-sample variability in terms of the number of mobile element insertions compared to other variant types.

As much as 97% of the annotated variations were located outside the previously known functional elements, including upstream, downstream, intron, and intergenic variants (Table 3). Of those found in exonic regions, most were synonymous, missense, or splice region variants. We discovered 4105 disruptive mutations, which were mostly frameshift (2453) and nonsense variants (1585). The median frequency of frameshifts and nonsense variants was 0.005 [IQR = 0.003–0.018] and 0.009 [IQR = 0.004–0.052], respectively. At the individual genome level, disruptive mutations were not common, with an average of 273 variants introducing or eliminating start or stop codons and 351 variants resulting in a frameshift.

### 2.4. Relation between Latvian and Global Populations

We performed a robust population structure analysis using the ADMIXTURE and principal component analysis (PCA), including a limited set of global and European samples from publicly available 1000 Genomes project (1000G) and Allen Ancient DNA Resource (AADR) datasets. As expected, the PCA displayed notable clustering of the Latvian population within the European cluster, though this was distinctly separate from other 1000G European sub-populations (Figure 1). ADMIXTURE analysis of K = 6 showed the lowest cross-validated error of 0.12813. Similar to PCA, ADMIXTURE shows clear distinction between geographically distant populations with noticeable heterogeneity within the population of Latvia (Figure 1). When put in the context of ancient genomes, the Latvian population approaches clusters of ancient European populations of Western-European hunter-gatherers and Yamnaya (Figure 1), with the age of the samples from Latvia mirroring gradual admixture events from major European source populations.

### 2.5. The Medical Relevance of Identified Variants

In our dataset, we detected 344 pathogenic variants according to the ClinVar database, with an average of 40 pathogenic variants per individual, while some scored up to 56 or as low as 26 (Table 3). As expected, the pathogenic variants appeared to be relatively rare, having a median frequency of 0.008 [IQR = 0.003–0.037]. The frequencies of pathogenic alleles within the Latvian cohort were similar to the total allele frequencies (AF) calculated for other populations represented in the gnomAD r2.1.1 exome dataset (*p*-value = 0.91, df = 567.67, t = 0.10), with some exceptions such as rs147574249 (AF in Latvian population = 0.02, AF in gnomADe = 0.10) or rs13222 (0.04, 0.0002). In addition, 43 protective variants were detected with an average of 18 variants per individual.

### 2.6. Imputation Panel Comparison

We merged the genome sequences of all 502 individuals into the Latvian population-specific reference panel (LVBMC) and compared its imputation performance with publicly available and ethnically heterogenous imputation reference panels: (1) the reference panel from a recent release of 1000G comprising WGS of 3202 individuals, (2) the same 1000G panel with relatives excluded resulting in 2504 unrelated individuals, and (3) the TOPMed R2 panel of 97,256 genomes (Figure 2). 

In total, the LVBMC panel imputed 12.1 million SNVs or 98.56% compared to the 1000G panel (Figure 2) and 3.96 million INDELs, which is 1.26 times more than the 1000G panel. The LVBMC panel performed better, particularly for SNVs with a frequency of 0.05% or higher, while the number of high-confidence low-frequency variants imputed with the LVBMC panel was the lowest among all of the panels tested. In total, 11.38 million imputed variants overlapped between the LVBMC and 1000G panel, with 9.4 million variants falling into the category of high-confidence genotypes (INFO score > 0.8). By using the LVBMC imputation reference panel (IRP) we managed to impute 4.68 × 10^6^ unique variants (43% SNVs), while 1000G imputed 128,081 (64.4% SNVs) variants that were not imputed by the LVBMC IRP. These uniquely imputed variants had an average frequency of 0.099 [IQR = 0.005–0.095] for the LVBMC and 0.036 [IQR = 0.002–0.008] for the 1000G panel-specific variants, respectively. We then aimed to investigate the impact of the presence of relatives in the IRP dataset on the imputation performance. Therefore, we developed another 1000G-derived dataset by excluding relatives from it, which decreased the number of imputed high-confidence SNVs by 2.1%. Finally, we tested the imputation performance of the TOPMed dataset, a large reference panel comprising 97,256 genomes, which provided 2.4% more SNVs in total, and 26.0% more high-confidence SNVs compared to the LVBMC IRP (Figure 2).

In order to evaluate the imputation accuracy of our panel, we performed a separate imputation analysis by simulating an array dataset consisting of 200 out of 502 overall sequenced samples. To simulate the array data, we randomly selected 500,000 SNPs to be imputed, allowing the rest of the SNPs to serve as a gold standard for concordance analysis. The dataset of 200 simulated array samples was then imputed using a reduced LVBMC IRP, consisting of 302 remaining genomes and 1000G IRP. In total, accuracy was assessed for 9.4 × 10^6^ variants imputed by the reduced LVBMC panel overlapping gold standard variants and 8.4 × 10^6^ variants imputed by the 1000G panel, respectively. Although the haplotype prediction was close to equal between both panels tested (reduced LVBMC and 1000G), accuracy was slightly lower for the reduced LVBMC IRP with a total concordance of 92.85% compared to 93.20% for the 1000G IRP (Table 4). Both panels had an equal NR sensitivity of 0.89, while NR discordance was lower for 1000G with 0.117 compared to 0.134 for the reduced LVBMC IRP. In addition to the imputation of the random simulated array, we also imputed only the WGS-derived variants that were overlapping Global screening array genotypes (230,451 in total); however, no significant change in the accuracy of imputed variants was observed.

## 3. Discussion

The representation of Latvian genomes in a large genomic dataset such as gnomAD is virtually absent. This fact significantly impacts genetic research globally and negatively influences the diagnostics of monogenic diseases and cancer. This is the first report on the genetic variance of the Latvian population, presenting results from the high-coverage whole-genome sequencing (WGS) of 502 individuals from the population of Latvia. We confidently called 18,266,684 small and 61,605 large variants with 601,374 polymorphisms being novel compared to gnomAD. We also constructed a population-specific imputation reference panel and showed that it can improve the accuracy and performance of imputation for common and low-frequency variants.

The use of large national biobanks to establish a population reference genome helps to avoid the bias that may arise from smaller, targeted (often disease-specific) cohorts as a source of individual genomes. In this study, there was no phenotype-based cohort selection performed before sequencing. Instead, we attempted to include all the available genomes from the Genome Database of the Latvian Population, keeping the basic inclusion criteria applied only to the quality of sequencing data. Nevertheless, the final study group appeared to be diverse, representing the Latvian population in terms of age, sex, ethnicity, and geographic distribution. The general assessment of WGS data showed high quality, with most of the genomes having an average of 30× coverage and almost 20% of sequences reaching 50× coverage. The quality of variant calling was similar to previously reported [15,16], showing a 98.2% concordance rate and 0.96 NR sensitivity when compared to array-based genotyping. Since the aim of this study was not to perform an in-depth characterization of population structure and ancestry markers, we did not attempt to include a larger set of other populations with available genotypes and restricted ourselves to the expanded 1000G dataset including 3202 samples and the Allen Ancient DNA Resource (AADR) dataset of 5981 ancient and modern samples. Overall, the principal component analysis (PCA) showed consistent results with previous regional studies [5,17], where the clustering of populations within the PCA mirrors their geographic distances. Nevertheless, an in-depth population analysis of a larger number of Latvian samples will be reported elsewhere. 

The overall variant diversity distribution in our study was consistent with those reported in other sequenced populations [15,17,18,19,20], with the main differences stemming from the total number of WGS samples included in the analysis. Sequencing of 1076 genomes in Poland with an average coverage of 35.3× resulted in 39.3 million small variants, while this number increased to 76 million in 1171 genome sequences from Brazil with an average coverage of 38.6× [19,20], compared to 30.9 million unfiltered variants discovered in our cohort. Compared to the mean of 3.8 million SNVs discovered in our samples, sequencing of genomes from Poland resulted in similar results of 3.71 million single-nucleotide variants per genome [18], two sequenced individuals from United Arab Emirates (UAE) yielded 3.9 and 4.0 million variants per genome [21], while 97 sequenced genomes in Ukraine resulted in the average of lower number of 3.48 million SNVs [17]. The number of discovered small insertions and deletions was also similar between studies. Compared to 0.65 million indels found in a Latvian cohort, a study in Poland found a mean of 0.7 million indels per genome, while 6.4 and 6.6 million indels were called in the two genomes from the UAE [18,20]. Unlike for SNVs where numbers were lower, the cohort from Ukraine showed an increased number of 1.48 million indels per individual [17].

The proportion of novel variants was 3.3%, though markedly higher for structural variants (41.78% novel) and polymorphisms affecting the reading frame of the gene (10.97% novel) or altering a protein structure (23.88% novel). It should be noted that the actual number of variants identified in our study was larger, and here we report only those variants that were identified on at least three occasions in the case of small variants (SNVs and INDELs). The particular criterion of the minimum number of variant occurrences in the analysed cohort was set to increase the reliability and ensure the quality of called variants for subsequent analysis, retaining 18.3 million SNVs and indels from a total set of >30 million variants identified. It is clear that such an approach significantly decreases the total number of rare and unique variants detected. Despite this, we managed to report 344 pathogenic or likely pathogenic variants in our dataset. The frequency of these clinically relevant variants was similar to gnomADe populations, with some exceptions, such as missense variant rs13222 in the *ARK1C2* gene from the aldo/keto reductase superfamily—involved in the conversion of aldehydes and ketones to their corresponding alcohol—which is highly expressed in fat and liver tissue, and rs147574249, also a missense variant within the *FCGR3B* gene—involved in immune regulation—with highest expression in the spleen and appendix [22]. More careful investigation is needed to assess the population-specific variants of medical relevance, considering all identified potentially pathogenic variants and including a manual inspection step to assess their quality.

Consistent with previous studies, our results show that the use of population-specific imputation reference panels significantly improves the genotype imputation [13]. Despite the relatively small sample size of the established LVBMC reference panel, the panel enhanced the imputation by an increased number of common (>5%) and low-frequency (0.5–5%) SNVs. Confidentially imputed variants that were unique to the LVBMC IRP showed high frequency at 19% compared to only 2% for variants imputed by the 1000 Genomes Project (1000G) panel. However, unlike shown in previous studies [13], we did not observe improvements in the imputation of rare variants (<0.5%). The poorer imputation performance of rare variants most likely arose from the relatively small reference panel sample size and also the resulting underrepresentation of rare alleles in our dataset compared to conventional imputation panels. In addition, it can be attributed to the exclusion of relatives and the applied phasing method, which did not consider linkage information from raw reads [13]. Meanwhile, the notably better performance of the 1000G imputation in the case of rare variants may be explained by the inclusion of 600 additional trios and higher coverage [15]. Importantly, we expect results to change considerably with the increase in the LVBMC IRP sample size.

The concordance between WGS data and array-derived genotypes was relatively high (98.5% for reference haplotypes) and consistent with previous studies [17]. Nevertheless, when focusing on the imputation accuracy, we did not observe improvement comparing variants imputed by the LVBMC and 1000G panels. Interestingly, although concordance with array genotypes was better than with imputed genotypes, both panels produced proportionally fewer false positive non-references, indicating possible limitations of WGS accuracy. Finally, even imputation based on a small panel consisting of 302 LVBMC genomes displayed 92.85% total concordance with actual genotypes in the independent set of 200 samples.

Results of our assessment as well as similar studies show that increased population-specific variation frequency and longer-shared high linkage disequilibrium (LD) regions of populations with local IRP enhance imputation and can result in detection of additional medically relevant polymorphisms [13,23]. Further research should evaluate if higher admixture of European founding populations in the LVBMC IRP might be beneficial for haplotype prediction in a broader cohort of European descent individuals as well as ancient genomes, due to the higher frequency of variants associated with ancient European populations. Additionally, pangenome reference could be constructed from existing WGS samples, further improving alignment and variant calling of genomes from the population of Latvia [24,25]. 

It is crucial to address the ethical aspects of this research, particularly when considering the extended use of population-based genome variation references. The current LGDB framework offers a robust solution for safeguarding personal data. This includes a government-regulated pseudonymization process and comprehensive informed consent, which permits the broader application of the data collected. Furthermore, we are proactively participating in the European ‘1+ Million Genomes’ Initiative to ensure enhanced cross-border access to population-specific genomic datasets. One of the study’s limitations is the relatively small sample size, as evidenced by the imputation performance on rare variants. Continued sequencing efforts are essential to address and enhance this situation.

In conclusion, we have presented as far as we know the first analysis of a significant number of genome sequences from the Latvian population. This high-quality dataset provides a detailed summary of various types of genetic variation and shows its potential to improve imputation. A Latvian population reference is an important asset for future population genetics research, as it fills an important gap in European genetic landscape.

## 4. Materials and Methods

### 4.1. Cohort

The study group was selected from The Genome Database of Latvian Population (LGDB) resources. LGDB ensures patient recruitment, blood sample collection, primary processing, and relevant anthropometric data according to previously developed standard procedures [26]. We selected 502 whole-genome sequences that passed the quality criteria (see Section 4.4) from the total of 887 participants of the LGDB for whom whole-genome sequencing (WGS) data were available in July 2022. A set of 316 (63%) samples from this cohort had additional quality-controlled genotype information, which was used for concordance analysis. For principal component analysis (PCA) and imputation, we included an expanded 1000G dataset [15], composed of 3202 phased GRCh38 samples and 117,175,809 small variants. Additionally, a TOPMed imputation panel was included consisting of 97,256 samples [27,28,29]. Finally, 5981 ancient and modern samples from the AADR V54.1.p1 (1240K + HO) [14] were included for population structure analysis, with source and additional information for each sample provided in Appendix A.

### 4.2. WGS Sequencing and Genotyping

For the DNA isolation, the phenol-chloroform extraction method on peripheral blood leukocytes was applied according to the LGDB standard procedures [26]. Both the genome-wide DNA library preparation and sequencing were performed in Latvia MGI Tech laboratory using the MGI automated sample processing and high-throughput genome sequencing platforms. The PCR-free DNA libraries were prepared with the MGIEasy PCR-Free DNA Library Prep Set (MGI Tech Co., Ltd., Wuhan, China) on MGISP-960 High-throughput Automated Sample Preparation System (MGI Tech Co., Ltd., Wuhan, China). The quantity and quality of the DNA libraries were evaluated using the Qubit fluorometer (ThermoFisher Scientific, Waltham, MA, USA) and a 2100 Bioanalyzer instrument (Agilent Technologies, Santa Clara, Ca, USA), respectively. The WGS was performed on the DNBSEQ-T10×4RS sequencing platform (MGI Tech Co., Ltd., Wuhan, China) using DNBSEQ-T10×4RS High-throughput Sequencing Set (FCL PE150) (MGI Tech Co., Ltd., Wuhan, China). DNBSEQ sequencing platforms have been previously benchmarked against other technologies, including Illumina-based instruments, showing comparable levels of sequencing quality, uniformity of coverage, percent GC coverage, variant accuracy, and providing the cost-efficient solution for WGS [30,31].

### 4.3. Variant Calling

Sequences were processed according to GATK best practices. Sequenced reads were trimmed with trim-galore [32] v0.6.7 and aligned to GRCh38 human reference genome using BWA-mem2 [33] v2.2.1, with mapped read quality assessed using bamQC [34]. Aligned reads were sorted with samtools [35] v1.9. We used GATK 4.2.6.1 [36] for further processing. Duplicates were marked with GATK MarkDuplicatesSpark, and base quality score recalibrated with dbSNP146 using GATK applyBQSR. Three main variant types were analysed separately. Firstly, small variants were called using GATK HaplotypeCaller with -ERC GVCF option enabled for further combined variant calling using GATK GenotypeGVCFs with the default options on 50 MB chunks. Secondly, structural variants were called with Manta [37] v1.6.0 and merged using SURVIVOR [38] v1.0.7. And lastly, mobile elements were called using MELT [39] v2.2.2. Parallel [40] v20220522 was used to distribute computation on Riga Technical University HPC cluster computers and Singularity [41] to install the necessary software. Nextflow v21.10.6.5660 was used to automate all performed analyses [42].

### 4.4. Quality Control and Annotation

Only samples with completed analysis of small variants, structural variants (SV), and mobile elements (MEI) were included. Additionally, closely related samples (PLINK2 pi_hat > 0.1875) and those with >3 SD deviation from the mean heterozygosity were excluded [13]. GATK VariantRecalibrator was used to keep variants with a truth sensitivity of 99.8% and bcftools [35] v1.9 was used to retain variants with a minimum allele count of three. No frequency filter was applied to structural variants and mobile elements; however, structural variants were filtered for false positives by removing variants with |SVLEN| outside the range of 50 to 1 × 10^7^. PLINK2 [43,44] v2.00a2.3 was used to exclude variants in linkage disequilibrium and to perform ADMIXTURE [45] and PCA [43] of all 502 individuals. Stratification between WGS batches was corrected by filtering out variants with >50% allele frequency mismatch between the cohorts, as described in Bergström et al. [46]. For the annotation of small variants, we divided multi-allelic entries into separate variants and annotated them using Ensembl VEP [47,48] 107.0, which includes ClinVar 202201 [48] and gnomAD r2.1.1 [49] datasets. Structural variants and mobile elements were annotated using AnnotSV [50] v3.1.2. To compare the distribution of allele frequencies, we used a two-sided two-sample chi-square test for equality of proportions with continuity correction using prop.test function from the statistics package in R [51] v4.2.0.

### 4.5. Phasing, Imputation, and Concordance Analysis

Imputation analysis followed Genotype imputation workflow v3.0 V.1 [52] with workflow flowchart shown in Appendix A. Eagle [16] v2.4.1 was used for phasing, while Beagle [53] v4.1 performed imputation with the “window = 500,000” option added. Latvian population-specific imputation panel (LVBMC) was created by merging all 502 samples, filtering for a minimum allele count of three, and converting to Bref format. Imputation was performed on a separate cohort of 200 quality-controlled array samples (Appendix A) genotyped using an Infinium Global Screening Array (Illumina, San Diego, CA, USA) on the iScan System microarray scanner (Illumina, San Diego, CA, USA).

To evaluate the accuracy of imputation, 302 individuals were randomly selected from the initial set of 502 samples, filtered for a minimum allele count of three, and converted to the Bref format to develop a reduced LVBMC panel (Appendix A). For the remaining 200 individuals, 500,000 variants were randomly selected to simulate array data to be imputed, while the rest of the variants were used as the gold standard for concordance analysis to evaluate the accuracy of imputed polymorphisms. Our method of assessing imputation accuracy differs from similar assessment by Mitt et al., 2017 [13], where separate whole-exome sequencing data was used as a gold standard, and Byrska-Bishop et al., 2022 [15], who used separate sequencing samples from extensive multi-generation sequencing efforts [54]. Concordance was tested using SnpSift [21] v4.3t software. Additionally, we calculated NR sensitivity and NR discordancy [13] for WGS samples with genotyped calls as the gold standard.

## Figures and Tables

**Figure 1 ijms-24-15345-f001:**
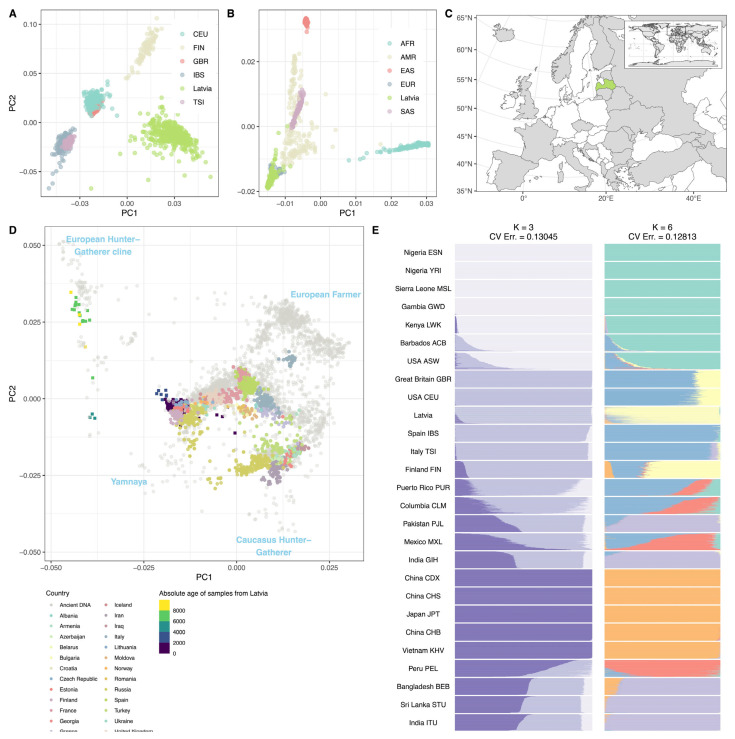
Principal component analysis (PCA) of merged LVBMC, 1000G, and AADR datasets. (**A**) European cluster with Latvian, Finnish (FIN), Central European (CEU), British (GBR), Spanish (IBS), and Italian (TSI) samples. (**B**) Zoomed-out global population clusters with European (EUR), African (AFR), American (AMR), East Asian (EAS), South-East Asian (SAS) groups. (**C**) Map of countries for modern genomes included in PCA with Latvia highlighted. (**D**) AADR ancient (grey circles) and modern European clusters (coloured circles). Samples from Latvia are represented by squares and coloured by age. Ancient clusters annotated according to Allentoft et al. 2022 [6]. Age defined as zero for modern and years before 1950 for ancient samples [14]. (**E**) ADMIXTURE analysis of 1000G and LVBMC WGS samples.

**Figure 2 ijms-24-15345-f002:**
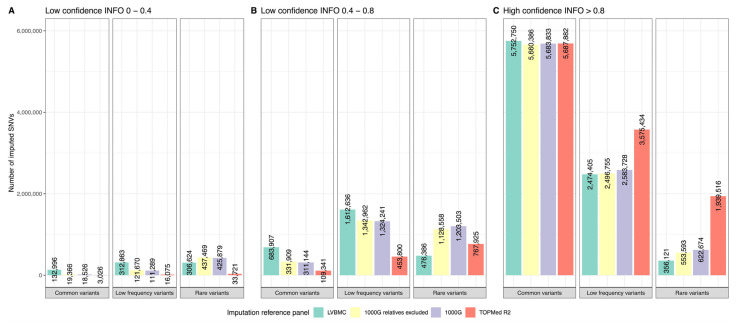
Number of SNVs imputed from different imputation reference panels (IRP): (**A**–**C**) Number of common (>5%), low-frequency (0.5–5%), and rare (<0.5%) variants imputed for Latvian array genotypes.

**Table 1 ijms-24-15345-t001:** Baseline description of study group included in the analysis.

	Female	Male	Total
	(N = 318)	(N = 184)	(N = 502)
**Age**			
Mean (SD)	54.4 (13.6)	50.0 (14.4)	52.8 (14.0)
Median [Min; Max]	55.0 [21.0; 91.0]	50.5 [18.0; 87.0]	54.0 [18.0; 91.0]
**Region**			
Courland	47 (14.8%)	32 (17.4%)	79 (15.7%)
Latgale	63 (19.8%)	34 (18.5%)	97 (19.3%)
Riga	53 (16.7%)	41 (22.3%)	94 (18.7%)
Semigallia	28 (8.8%)	8 (4.3%)	36 (7.2%)
Vidzeme	74 (23.3%)	49 (26.6%)	123 (24.5%)
Missing	53 (16.7%)	20 (10.9%)	73 (14.5%)
**Ethnicity**			
Belarussian	9 (2.8%)	4 (2.2%)	13 (2.6%)
Latvian	192 (60.4%)	124 (67.4%)	316 (62.9%)
Polish	4 (1.3%)	3 (1.6%)	7 (1.4%)
Russian	90 (28.3%)	43 (23.4%)	133 (26.5%)
Ukrainian	13 (4.1%)	3 (1.6%)	16 (3.2%)
Missing	10 (3.1%)	7 (3.8%)	17 (3.4%)

**Table 2 ijms-24-15345-t002:** Variation summary of 502 Latvian whole-genome sequences grouped by type.

Variant Type	Total Count	Novel gnomAD	Mean (SD [Min; Max])	Mean Novel (SD [Min; Max])
Small variants *				
SNV	15,158,133	466,749	3,822,725 (74,939 [3,643,058; 4,022,661])	9788 (4938 [4386; 23,979])
Deletion	1,608,579	49,249	326,497 (7337 [304,340; 349,310])	1444 (278 [895; 2478])
Inframe deletion	1398	76	203 (14 [160; 292])	4 (3 [1; 56])
Insertion	1,435,886	59,499	327,355 (7352 [305,045; 349,737])	2399 (327 [1686; 5622])
Inframe insertion	1083	61	176 (13 [138; 301])	3 (4 [1;67])
Structural variation **				
DEL	28,191	10,337	3465 (244 [2568; 4179])	794 (88 [514; 1067])
DUP	6866	3139	456 (71 [256; 668])	192 (29 [104; 277])
INS	10,405	5791	798 (345 [327; 1993])	454 (198 [196; 1135])
INV	3881	1534	297 (36 [179; 396])	134 (17 [84; 184])
TRA	7431	3980	514 (99 [379; 796])	292 (56 [208; 438])
Mobile element insertions **				
ALU	4074	760	183 (108 [30; 504])	19 (11 [1; 61])
LINE1	528	141	19 (15 [1; 74])	7 (4 [1; 24])
SVA	207	49	7 (6 [1; 27])	2 (2 [1; 9])
HERVK	22	9	1 (0 [1; 4])	1 (0 [1; 2])

* Allele count ≥ 3; ** Allele count ≥ 1.

**Table 3 ijms-24-15345-t003:** Summary of variant annotation of 502 Latvian whole-genome sequences.

Variant Type	Total Count	Novel gnomAD	Mean (SD [Min; Max])	Mean Novel (SD [Min; Max])
Variants by location				
Upstream gene variant	2,972,390	76,822	713,153 (25,203 [663,145; 775,844])	1924 (1428 [736; 6062])
5 prime UTR variant	58,753	4215	12,434 (432 [11,331; 14,018])	90 (117 [9; 531])
Noncoding transcript exon variant	504,846	13,166	123,426 (5194 [113,905; 134,625])	316 (309 [82; 1210])
3 prime UTR variant	237,749	5596	57,344 (1625 [54,020; 61,640])	127 (120 [28; 472])
Downstream gene variant	3,112,542	71,227	757,570 (26,960 [704,675; 821,758])	1818 (1259 [722; 5243])
Intergenic variant	7,374,606	362,260	1,651,845 (25,220 [1,564,267; 1,736,263])	9031 (1648 [5484; 14,930])
Intron variant	10,367,399	194,729	2,634,492 (48,769 [2,512,167; 2,764,304])	4368 (3055 [1751; 12,885])
Functional variants by type				
Splice acceptor variant	1319	66	269 (13 [232; 307])	3 (1 [1; 13])
Splice donor variant	1855	70	393 (16 [351; 449])	6 (2 [2; 16])
Splice-donor-region variant	3471	123	839 (45 [754; 953])	3 (3 [1; 15])
Splice donor fifth-base variant	1499	73	341 (17 [299; 389])	5 (2 [1; 12])
Splice-region variant	19,629	254	4862 (179 [4603; 5262])	5 (5 [1; 36])
Transcript ablation	4	0	2 (1 [1; 3])	0
Frameshift variant	2453	269	351 (66 [273; 980])	11 (21 [1; 335])
Missense variant	67,397	3617	12,257 (1176 [10,682; 14,876])	75 (112 [1; 421])
Start lost	234	10	44 (5 [29; 60])	1 (1 [1; 6])
Stop gained	1087	28	152 (18 [122; 203])	1 (2 [1; 28])
Stop lost	264	9	77 (7 [58; 98])	1 (0 [1; 3])
Synonymous variant	51,160	1180	12,225 (706 [11,127; 13,741])	27 (35 [1; 140])
Protein-altering variant	67	16	4 (5 [1; 90])	2 (5 [1; 44])
Medically relevant variants in the Latvian population				
Pathogenic	344	-	40 (5 [26; 56])	-
Likely pathogenic	177	-	19 (3 [11; 30])	-
Benign	93,468	-	35,285 (575 [33,879; 36,871])	-
Likely benign	19,519	-	1412 (58 [1253; 1617])	-
Protective	43	-	18 (3 [9; 26])	-
Drug response	203	-	76 (9 [53; 106])	-
Association	183	-	81 (8 [58; 101])	-

**Table 4 ijms-24-15345-t004:** Comparison of whole-genome sequencing-derived genotypes determined by array or imputed using 1000G and reduced Latvian population-specific reference panels.

		WGS Genotype Overlap (% [Mean; SD (Min; Max)])		
Compared Set		REF	ALT_1	ALT_2
Array (n = 316)	REF	**98.5** [318,418; 2491 (309,942; 324,671)]	0.2 [219; 25 (167; 424)]	0.2 [135; 162 (1; 642)]
Imputed with 1000G panel (n = 200)	REF	**96.4** [4,976,995; 31,027 (4,839,014; 5,030,586)]	10.8 [211,324; 9452 (193,120; 277,490)]	1.0 [13,574; 1339 (10,398; 20,400)]
Imputed with reduced LVBMC panel (n = 200) *	REF	**96.6** [5,724,931; 59,377 (5,532,919; 5,811,414)]	12.8 [275,223; 20,029 (238,691; 354,592)]	1.8 [25,167; 2280 (20,140; 34,645)]
Array (n = 316)	ALT_1	0.9 [3067; 1901 (702; 6486)]	**99.5** [105,725; 3716 (99,028; 113,350)]	0.5 [281; 180 (46; 955)]
Imputed with 1000G panel (n = 200)	ALT_1	3.2 [165,345; 11,462 (140,740; 219,026)]	**85.1** [1,671,734; 28,554 (1,583,691; 1,739,678)]	6.0 [78,923; 3714 (72,727; 104,440)]
Imputed with reduced LVBMC panel (n = 200) *	ALT_1	3.2 [188,676; 16,716 (158,304; 280,839)]	**83.0** [1,784,775; 37,177 (1,669,064; 1,868,066)]	6.0 [82,413; 5783 (73,026; 117,744)]
Array (n = 316)	ALT_2	0.6 [1781; 654 (430; 3442)]	0.3 [347; 38 (273; 672)]	**99.3** [59,945; 1873 (56,492; 65,564)]
Imputed with 1000G panel (n = 200)	ALT_2	0.4 [21,463; 5702 (9101; 33,309)]	4.2 [81,625; 4195 (73,042; 111,549)]	**92.9** [1,214,036; 19,902 (1,150,271; 1,261,170)]
Imputed with reduced LVBMC panel (n = 200) *	ALT_2	0.2 [13,949; 2369 (9743; 23,925)]	4.2 [90,491; 6953 (77,116; 133,932)]	**92.2** [1,268,448; 21,304 (1,189,978; 1,317,623)]

REF—Homozygous reference 0/0 haplotype; ALT_1—Heterozygous variant 0/1 haplotype; ALT_2—Homozygous variant 1/1 haplotype; * The reduced LVBMC imputation reference panel is a subset of the initial LVBMC reference panel consisting of 302 instead of 502 genomes. The percentage indicates the average proportion of haplotypes matching (e.g., REF/REF) or mismatching (e.g., REF/ALT_1) for one individual. Bold highlights figures indicating concordance between genotypes in each set.

## Data Availability

All data used in this study can be requested from the Latvian Genome Database. Individual-level sequence datasets (FASTQ files) and variant calling datasets (VCF) are deposited in the European Genome-phenome Archive (EGA), which is hosted by the EBI and the CRG, under EGA study accession number EGAS00001007406. Analysis code pipelines for replicating the study results are available on GitHub at https://github.com/raimondsre/genome_reference_Latvia (accessed on 6 September 2023).

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
