# Peer review of "Whole-Genome Sequencing of 502 Individuals from Latvia: The First Step towards a Population-Specific Reference of Genetic Variation"

_ijms, 2023, doi:10.3390/ijms242015345_

Round 1
Reviewer 1 Report
This study aims to address a significant gap in the field of genomics by focusing on the Latvian population. The paper describes variation in this population with unique genetic ancestry derived from Western European hunter-gatherers and the Yamnaya culture developing the population-scale reference database of Latvian genetic variation may fill the gap in the map of genomic diversity in Europe and improve genomics research globally. While the paper does an excellent job outlining the current state of human population genetics and the gaps in data representation, it could be enriched by incorporating historical or anthropological context to provide deeper insight into Latvia's unique genetic features. The paper is generally well-written and easy to understand. Perhaps, pointing at the interdisciplinary applications of the research in the Introduction could broaden its appeal to a wider scientific audience.
A few comments and clarifications classified by manuscript sections are listed below:
Methodology
1. Sampling Scale: The study employs Whole Genome Sequencing (WGS) on a dataset comprising 502 Latvian individuals. It raises the question of whether sequencing more individuals would offer additional insights into genomic diversity.
2. Population Representation: Further clarification on the demographic composition of the sample—perhaps including a map of the sample collection—would enhance the study's validity.
3. Quality Criteria: The study uses specific criteria to include genomes for analysis. The potential biases introduced by these criteria should be addressed. Specifically, please clarify what “closely related samples” meant in the exclusion criteria. The paper states that “Only samples with a full set of called small variants, structural variants (SV), and mobile elements (MEI) were included" is somewhat unclear. One of each? At least 100 of each? What does the "full set" mean at this step?
4. Sequencing Platform: The DNBSEQ-T10×4RS sequencing platform is employed, but a justification for this choice is lacking. Information on its reliability or accuracy would be useful.
5. Software Version: Providing the version number for GATK would add to the paper's reproducibility. GATK version is listed only for a single sub-tool (DuplicatesSpark). If different versions of GATK components were used in the pipeline - it has to be clarified. If only v4.2.6.1 was used -it has to be explicitly stated.
Results and Interpretation
1. Variant Identification: The study claims to have identified 18.2 million variants, with 3.3% being novel. An assessment of the statistical validity of these results is warranted. The manuscript also states "In total, 30,919,589 SNPs and INDELs were detected with a false discovery rate of less than 0.01%"_How was this false discovery rate assessed?
2. Imputation Accuracy: The section on imputation accuracy is incomplete and could benefit from further details, especially regarding the measurement techniques employed.
3. Statistical Tests: The paper claims that the "LVBMC panel performed better, particularly for SNVs with a frequency of 0.05% or higher." And the number of high confidence low frequency variants was the lowest among all the panels tested. Why is this the case? Does it mean other panels with large sample sizes are better at imputing rare variants than sizeable regional panels? Information on the statistical tests used to reach this conclusion would bolster the claim.
4. Concordance Metrics: The paper reports an "overall concordance of 98.81%" between sequencing and genotyping platforms. Consideration of additional metrics like sensitivity, specificity, or F1 scores could enrich the interpretation.
5. Admixture/Structure: The paper would greatly benefit from estimation of individual ancestries (ADMXITURE\STRUCTURE), and similar papers have presented these results before. Is there any reason this was not done in this study?
Discussion
1. Broader Impact: The paper argues that their Latvian-specific IRP could have broader implications in the fields of precision medicine and human genomics. Further elucidation on this point would be advantageous.
2. Ethical Considerations: Any ethical aspects related to this genomic study on the Latvian population should be openly discussed.
3. Comparative Analysis: Providing additional comparisons of the Latvian genomic data with other populations, both within Europe and globally, would add value. Recommendations on further population sequencing would be beneficial.
Author Response
Methodology
- Sampling Scale: The study employs Whole Genome Sequencing (WGS) on a dataset comprising 502 Latvian individuals. It raises the question of whether sequencing more individuals would offer additional insights into genomic diversity.
Thank you, we agree that the number of sequenced samples is a certain limitation of this study and therefore include the description of the limitations in the discussion section: “One of the study’s limitations is the relatively small sample size, as evidenced by the imputation performance on rare variants. Continued sequencing efforts are essential to address and enhance this situation”. Nevertheless due to the scarcity of genomic data from Latvia we believe that reporting the currently available genomic information is crucial to promote its use for further research.
- Population Representation: Further clarification on the demographic composition of the sample—perhaps including a map of the sample collection—would enhance the study's validity.
We appreciate your suggestion, unfortunately for the data protection purposes LGDB questionnaires do not allow to discriminate the exact place of birth. As we have a rather centralized sample collection this would also not represent the origin of participants. To clarify this situation we have modified the sentence in the Results section: “The regional distribution of the study participants according to their self-reported place of birth, (which is restricted to indicating one of the five regions of Latvia without specifying the exact birthplace), self-reported ethnicity, as well as their age and sex, is provided in Table 1.” . We also included a self reported region of origin in table 1.
- Quality Criteria: The study uses specific criteria to include genomes for analysis. The potential biases introduced by these criteria should be addressed. Specifically, please clarify what “closely related samples” meant in the exclusion criteria. The paper states that “Only samples with a full set of called small variants, structural variants (SV), and mobile elements (MEI) were included" is somewhat unclear. One of each? At least 100 of each? What does the "full set" mean at this step?
Thank you for pointing out the missing clarification, we have supplemented methods section with relevant metrics. In order to filter out related individuals we used PLINK pi-hat score above 0.1875 removing first degree cousins. Additionally, we clarify variant calling, which was performed in three separate pipelines. Only samples with completed analysis of all three variant types (full set) were included, which is further clarified in the results section statement “295 sequences were not further processed due to missing data for structural variants and mobile element insertions”.
- Sequencing Platform: The DNBSEQ-T10×4RS sequencing platform is employed, but a justification for this choice is lacking. Information on its reliability or accuracy would be useful.
We used DNBSEQ-T10×4RS platform as the lowest cost sequencing solution. We included additional explanation in the Methods section providing references to the previously published benchmarking exercises showing equal sequencing quality to the most used Illumina technology. Additionally, our own quality assessment (Table 4) comparing concordance to genotyped samples shows similar results.
- Software Version: Providing the version number for GATK would add to the paper's reproducibility. GATK version is listed only for a single sub-tool (DuplicatesSpark). If different versions of GATK components were used in the pipeline - it has to be clarified. If only v4.2.6.1 was used -it has to be explicitly stated.
Thank you for the suggestion, we have added clarification in the methods section on GATK v4.2.6.1 to reduce the ambiguity.
Results and Interpretation
- Variant Identification: The study claims to have identified 18.2 million variants, with 3.3% being novel. An assessment of the statistical validity of these results is warranted. The manuscript also states "In total, 30,919,589 SNPs and INDELs were detected with a false discovery rate of less than 0.01%"_How was this false discovery rate assessed?
Thank you for your valuable comment. We detected 30e6 small variants with GATK haplotype caller default Phred cutoff of 30, which translates to individual variant error rate (not FDR) of 0.1% (not 0.01%). We did not assess FDR empirically thus manuscript was corrected accordingly. In total 18.2 million variants passed quality control that was based on GATK best practices and similar assessments of WGS calling quality and implementation (Mitt et al. 2017) and variation diversity (Oleksyk et al. 2021). Most significant factor affecting the number of identified variants is exclusion of variants with allele count of 2 or less. Such quality criteria was implemented by Mitt et al. 2017 and other WGS studies (e.g. Byrska-Bishop et al. 2022), but not by Oleksyk et al. 2021 due to smaller sample size (97 WGS samples).
- Imputation Accuracy: The section on imputation accuracy is incomplete and could benefit from further details, especially regarding the measurement techniques employed.
Thank you for your suggestion, we have clarified the limitations of assessment of imputation accuracy in methods sections as well as added more details from accuracy assessment comparison between reduced LVBMC and 1000G IRP.
- Statistical Tests: The paper claims that the "LVBMC panel performed better, particularly for SNVs with a frequency of 0.05% or higher." And the number of high confidence low frequency variants was the lowest among all the panels tested. Why is this the case? Does it mean other panels with large sample sizes are better at imputing rare variants than sizeable regional panels? Information on the statistical tests used to reach this conclusion would bolster the claim.
Thank you for pointing out this ambiguity, we have clarified the section in the manuscript. Sentence “LVBMC panel performed better, particularly for SNVs with a frequency of 0.05% or higher” refers to the total number of variants, including low quality INFO <0.8. And indeed we show that the current imputation panel size of 502 samples is not enough to improve imputation of rare variants. For this comparison we used techniques implemented by Mitt et al. 2017 and other research evaluating imputation panels (such as Byrska-Bishop et al. 2022), mainly comparing imputation confidence assessments (INFO-value) in different frequencies of variants. However, higher INFO score does not guarantee correct inference, thus, we also performed concordance analysis of variants imputed by reduced LVBMC imputation panel of 302 samples (Table 4), approach that is further illustrated in schematic in Supplementary Figure 1.
- Concordance Metrics: The paper reports an "overall concordance of 98.81%" between sequencing and genotyping platforms. Consideration of additional metrics like sensitivity, specificity, or F1 scores could enrich the interpretation.
Thank you for your valuable suggestion, we have added non-reference sensitivity and non-reference discordancy values according to WGS evaluation by Mitt et al. 2017, treating genotyped calls as the gold standard for these particular metrics.
- Admixture/Structure: The paper would greatly benefit from estimation of individual ancestries (ADMXITURE\STRUCTURE), and similar papers have presented these results before. Is there any reason this was not done in this study?
Thank you for the suggestion, we have added a global ADMIXTURE plot as Figure 1E.
Discussion
- Broader Impact: The paper argues that their Latvian-specific IRP could have broader implications in the fields of precision medicine and human genomics. Further elucidation on this point would be advantageous.
Thank you for pointing it out, we have complemented the discussion section with a discussion on broader implications of the Latvian-specific IRP.
- Ethical Considerations: Any ethical aspects related to this genomic study on the Latvian population should be openly discussed.
Thank you for this valuable suggestion. We included the explanation for ethical considerations of this study in the discussion section:
“It's crucial to address the ethical aspects of this research, particularly when considering the extended use of population-based genome variation references. The current LGDB framework [VIGDB publikācija] offers a robust solution for safeguarding personal data. This includes a government-regulated pseudonymization process and comprehensive informed consent, which permits the broader application of the data collected. Furthermore, we are proactively participating in the European '1+ Million Genomes' Initiative to ensure enhanced cross-border access to population-specific genomic datasets.”
- Comparative Analysis: Providing additional comparisons of the Latvian genomic data with other populations, both within Europe and globally, would add value. Recommendations on further population sequencing would be beneficial.
Thank you for you comment, we have compared our results with multiple similar efforts across the world, including Brazil (Naslavsky et al. 2022), Ukraine (Oleksyk et al. 2021), Saudi Arabia (AlSafar et al. 2019) and Poland (Kaja et al. 2022). We have added a section of additional comparisons, however, it has to be kept in mind that these cohorts are both significantly larger and smaller than ours.
Reviewer 2 Report
The authors analyzed a comprehensive WGS dataset consisting of 502 individuals of Latvian descent sourced from the Genome Database of the Latvian Population. The study have established a regional reference database for Latvian genetic variations and observed a distinctive clustering pattern within the European subpopulations. The findings highlighted a notable Western-European hunter-gatherer and Yamnaya genetic heritage from Latvians. The study design is appropriate and the findings are impactful. However, I found some minor issues that can be improved to clarify the result presenting.
1. It would be great if the ages of Latvian samples legend of Fig 1C can be better presented. For instance, is the "2500" here means the years of the ancient samples to date from now?
Author Response
- It would be great if the ages of Latvian samples legend of Fig 1C can be better presented. For instance, is the "2500" here means the years of the ancient samples to date from now?
Thank you for your valuable comment, we will add the definition of age in the methods section of our manuscript in addition to the improved age scale in Fig 1C. In this case, age of ancient AADR samples is defined as Date mean in BP in years before 1950 CE [OxCal mu for a direct radiocarbon date, and average of range for a contextual date] (Mallick et al. 2022), while age of modern samples defined as participants biological age or zero in case of modern samples from AADR and 1000G dataset.
Reviewer 3 Report
Dear authors
thank you for the manuscript - this is interesting work collecting together and describing genetic variation in a specific population; I agree that your work should be interesting to scientists working in human genetics. I think the work is sound, and have only relatively minor suggestions.
1) maybe briefly describe the two ancestral populations, the Western European hunter-gatherers, and especially the Yamnaya population
2) line 22: maybe replace "genomics research globally" with "human genomics research"
3) line 27: maybe rephrase or clarify "notable though distinct"
4) line 38: Human population genetics "benefitted from" rather than "was based on" ?
5) line 266: maybe add "as far as we know" to your claim
6) line 270: maybe clarify "global anestry maximum"
7) can your data be described as a "pan-genome" ? if not, could you clarify the difference between your data and a pan-genome ?
8) are there no other publictations of population-specific reference genetic variation for other relatively well-defined populations?
no comments - the quality is fine
Author Response
1) Maybe briefly describe the two ancestral populations, the Western European hunter-gatherers, and especially the Yamnaya population.
Thank you for the suggestion. We have enhanced the introduction by adding a paragraph that briefly delves into the population genomics of Latvia.
2) Line 22: maybe replace "genomics research globally" with "human genomics research".
We added this suggestion to the manuscript, thank you.
3) Line 27: maybe rephrase or clarify "notable though distinct".
The clustering of the Latvian cohort within the European subpopulations is significant and easy to observe (notable), but at the same time, it is different or separate in some way (distinct).
4) Line 38: Human population genetics "benefitted from" rather than "was based on" ?
We added this suggestion to the manuscript, thank you.
5) Line 266: maybe add "as far as we know" to your claim
We added this suggestion to the manuscript, thank you.
6) Line 270: maybe clarify "global ancestry maximum"
Research incorporating ADMIXTURE analysis has shown European genomes to contain three major ancestral components, corresponding to ancestry of three distinct archaeological cultures. Western-European Hunter Gatherer shows highest ancestry proportion in Baltics (Mittnik et al. 2018) and, based on modern PCA analysis, we further extrapolate Latvian population having the highest EHG ancestral proportion among Baltic states, and thus exhibiting global maxima of one of the major ancestral components in Europe and globally.
However, without additional research..
7) Can your data be described as a "pan-genome"? If not, could you clarify the difference between your data and a pan-genome?
Our genome reference differs from pangenome graph reference. It only incorporates single nucleotide and small indel deviations from GRCH38 and cannot extrapolate more complex variation or be used as a base for an alignment. Although our dataset of 502 WGS samples could be assembled into regional pangenome reference, it’s a more complex analysis that was outside the scope of this study. We are aware of recent publications Liao et al. 2023 and Gao et al. 2023 that attempted to capture variation diversity beyond established alignment reference GRCH38, and we believe genomes of Latvian population might be proven beneficial to further improve impressive alignment and variant calling benefits provided by human pangenome references.
8) Are there no other publications of population-specific reference genetic variation for other relatively well-defined populations?
Thank you for this comment. To evaluate our findings, we compared results with multiple other population-specific references of genome variation, including Estonia, Ukraine, Brazil, Saudi Arabia and Poland. We supplemented the discussion section with additional details.